# Socioeconomic Deprivation and Invasive Breast Cancer Incidence by Stage at Diagnosis: A Possible Explanation to the Breast Cancer Social Paradox

**DOI:** 10.3390/cancers16091701

**Published:** 2024-04-27

**Authors:** Giulio Borghi, Claire Delacôte, Solenne Delacour-Billon, Stéphanie Ayrault-Piault, Tienhan Sandrine Dabakuyo-Yonli, Patricia Delafosse, Anne-Sophie Woronoff, Brigitte Trétarre, Florence Molinié, Anne Cowppli-Bony

**Affiliations:** 1Loire-Atlantique/Vendée Cancer Registry, 44093 Nantes, France; 2SIRIC ILIAD INCa-DGOS-INSERM-ITMO Cancer_18011, CHU Nantes, 44000 Nantes, France; 3French Network of Cancer Registries (FRANCIM), 31000 Toulouse, France; 4Côte d’Or Breast and Gynaecologic Cancer Registry, INSERM U1231, 21000 Dijon, France; 5Isère Cancer Registry, 38000 Grenoble, France; 6Doubs Cancer Registry, 25000 Besançon, France; 7Hérault Cancer Registry, 34000 Montpellier, France; 8EQUITY Research Team (Certified by the French League Against Cancer), CERPOP, UMR 1295, Université Toulouse III Paul Sabatier, 31000 Toulouse, France

**Keywords:** breast cancer, incidence, stage at diagnosis, socioeconomic deprivation, rurality

## Abstract

**Simple Summary:**

Compared to socioeconomically affluent areas, lower breast cancer incidence and similar or higher mortality have been reported in deprived areas. We provide a possible explanation to this paradox, by estimating and comparing incidence by stage at diagnosis between different levels of area-based socioeconomic deprivation in France. As deprivation increased, all-stages and early (stage I and stage II) invasive breast cancer incidence significantly decreased, while advanced (stage III–IV) breast cancer incidence significantly increased.

**Abstract:**

In this study, we assessed the influence of area-based socioeconomic deprivation on the incidence of invasive breast cancer (BC) in France, according to stage at diagnosis. All women from six mainland French departments, aged 15+ years, and diagnosed with a primary invasive breast carcinoma between 2008 and 2015 were included (*n* = 33,298). Area-based socioeconomic deprivation was determined using the French version of the European Deprivation Index. Age-standardized incidence rates (ASIR) by socioeconomic deprivation and stage at diagnosis were compared estimating incidence rate ratios (IRRs) adjusted for age at diagnosis and rurality of residence. Compared to the most affluent areas, significantly lower IRRs were found in the most deprived areas for all-stages (0.85, 95% CI 0.81–0.89), stage I (0.77, 95% CI 0.72–0.82), and stage II (0.84, 95% CI 0.78–0.90). On the contrary, for stages III–IV, significantly higher IRRs (1.18, 95% CI 1.08–1.29) were found in the most deprived areas. These findings provide a possible explanation to similar or higher mortality rates, despite overall lower incidence rates, observed in women living in more deprived areas when compared to their affluent counterparts. Socioeconomic inequalities in access to healthcare services, including screening, could be plausible explanations for this phenomenon, underlying the need for further research.

## 1. Introduction

While a clear socioeconomic pattern has been observed for most cancer types, with people from lower socioeconomic groups mostly affected, a paradox has been noted in the specific case of breast cancer (BC) [1]. Women living in socioeconomically deprived areas have been reported to present lower BC incidence rates and, simultaneously, similar or even higher mortality rates, when compared to their affluent counterparts [2]. This paradox could find its roots in differences between socioeconomic groups in the exposure to BC risk factors (lifestyle, reproductive history, and anthropometric factors) when considering overall incidence, and in access to and use of screening and other healthcare services, when considering mortality [3,4,5].

France has a universal healthcare system, and since 2004, its organized BC screening program has offered free access to a biannual mammography to every woman aged 50–74 years old (y/o) nationwide. Studies exploring socioeconomic disparities in organized screening participation in mainland France have found an inverted U-curve, with the highest participation rates being reported in areas with an intermediate deprivation level [6]. Nevertheless, in the Netherlands, where BC screening also targets 50–74 y/o women, socioeconomic inequalities in screening uptake have been related to a higher proportion of advanced BC and lower survival rates among people with low socioeconomic status [7]. Theoretically, these factors should translate to a higher incidence of advanced BC in lower socioeconomic groups, when compared to the higher ones. As a matter of fact, a study from Denmark comparing stage-specific BC incidence according to socioeconomic status, has previously confirmed this hypothesis in women aged 50+ y/o [8]. A higher incidence of distant stage BC was also found in young women (30–48 y/o) with a low income living in Norway when compared to their high-income counterparts, hypothesizing that low BC awareness and inequitable healthcare could be possible concurrent explanatory factors [9]. These findings could provide a possible explanation for the breast cancer paradox, as a more advanced stage at diagnosis is related to higher lethality and lower survival [10]. Nonetheless, further research, including a population with a wider age range and living in a country with universal health coverage, is needed to better understand the effect of socioeconomic deprivation on stage-specific BC incidence.

We previously explored this phenomenon in a study covering all women aged 18+ y/o from two mainland French departments (Loire-Atlantique and Vendée), where we reported a significantly reduced early (in situ and stage I) BC incidence in women living in (affluent or deprived) rural and deprived urban areas when compared to women living in affluent urban areas, but no significant effect of socioeconomic background on stages II–IV BC incidence [11]. However, in these two departments, people from affluent areas were overrepresented when compared to national standards, and participation rates at organized BC screening programs were among the highest in France [12]. These two factors together impacted the representativeness of the national French population. Moreover, a composite variable was used to explore the intrinsic relationship between rurality of residence and area-based socioeconomic deprivation, hindering the estimation of the specific effect of each factor. Finally, a less aggregated grouping of stages at diagnosis could have been useful to further understand this phenomenon. For these reasons, we considered it appropriate to extend the previous research to four mainland French departments (Isère, Doubs, Hérault, and Côte d’Or), which are also covered by Cancer Registries, in order to have a larger population and a better representation of French socioeconomic deprivation. Thus, this study included six French Departments, and our aim was to explore the influence of area-based socioeconomic deprivation on stage-specific invasive BC incidence.

## 2. Materials and Methods

### 2.1. Population

We included all women aged 15+ y/o diagnosed with a primary invasive breast carcinoma (excluding lymphoma and sarcoma) between 2008 and 2015, living at diagnosis in six mainland French departments (Loire-Atlantique, Vendée, Isère, Doubs, Côte d’Or, Hérault). They covered approximately 7.2 million people (11.1% of the mainland French population). All women with a previous diagnosis of in situ breast carcinoma and/or a Paget’s disease of the breast alone were excluded.

### 2.2. Data

Data on BC at diagnosis were provided by population-based cancer registries. The quality and completeness standards of these registries are certified every five years by the National Registry Evaluation Committee. Demographic and cancer variables included age and exact residential address at diagnosis, mode of detection (symptoms, organized screening, opportunistic screening), and stage at diagnosis (based on TNM classification of malignant tumors, 7th edition) [13]. The pathologic (pTNM) classification system was used if surgery was the first treatment, while the clinical (cTNM) classification was adopted in case of neoadjuvant or non-surgical treatment. Stages I and II were considered individually, while stages III and IV were grouped together to increase statistical power.

The 2011 French version of the European Deprivation Index (F-EDI), an area-based socioeconomic index, was used to measure individual socioeconomic deprivation. This index was constructed in a multi-step approach which included: (i) identifying the fundamental needs of people through a European survey specifically designed for this purpose; (ii) selecting the needs associated with both subjective and objective poverty; (iii) defining, using the selected needs, dichotomous individual deprivation indicators; (iv) selecting variables available both at the individual level and in the French census; and (v) constructing the F-EDI using the variables that were statistically associated with deprivation indicators [14]. The 10 variables retained measured overcrowding, access to heating, house ownership, unemployment, nationality, access to a car, working skills, number of people per household, level of education, and family composition (parents per household). The residential address of women diagnosed with a BC was geocoded using Geographic Information Systems (ArcGIS 10.2, ESRI, Redlands, CA, USA) and allocated to an IRIS (Ilots Regroupés pour l’Information Statistique) by the French national methodological platform for the study and reduction of health social inequalities in oncology (ERISC/MapInMed) [15]. An IRIS is the smallest geographic unit available in France defined by the French National Statistical Institute (INSEE) and contains 2000 people on average with relative homogeneous socioeconomic characteristics (France includes approximately 50,000 IRIS). Each IRIS was linked by the platform ERISC/MapInMed to a F-EDI score and a F-EDI quintile. F-EDI quintiles were defined by the platform according to the French national distribution of F-EDI score across different IRIS, ranging from Q1 for the most affluent area to Q5 for the most deprived area. This process allowed to attribute an F-EDI quintile to the BC cases included in the study. The women’s address was also employed to determine the rurality of residence, which was calculated on a municipal scale by INSEE [16]. The choice to include this last variable came from the fact that it is related to the territorial distribution of healthcare facilities, as well as to access to those facilities, and from the existing literature suggesting lower BC incidence rates in these territories [17]. Moreover, it is important to consider rurality of residence when assessing the effect of socioeconomic deprivation using the F-EDI, as this index fails to take the rural/urban context into account and as it enables exploration of other social and territorial dimensions that might interact with socioeconomic deprivation [14].

A complete case approach was followed, excluding cases with missing stage at diagnosis or F-EDI. From an initial population of 34,714 BC, a total of 33,298 BC (95.9%) and an at-risk population of 17,792,968 person-years were included in the analyses following exclusion of 1416 cases (4.1%), which had missing data relating to the F-EDI and non-geocoded addresses (323 BC; 0.9%), to the stage at diagnosis (1088 BC; 3.1%), or both variables (5 BC; 0.01%). The characteristics of BC cases excluded from the study according to exclusion criteria are presented in Appendix A.

### 2.3. Statistical Analysis

The characteristics of the study population were presented using percentages and compared between different F-EDI quintiles using Chi-squared tests (α-risk set at 5%). BC age-standardized incidence rates (ASIR) per 100,000 person-years were estimated and reported by age group (based on the organized screening program: 15–49, 50–74, and 75+ y/o), stage at diagnosis (all-stage, I, II, III–IV), F-EDI quintiles (from Q1 to Q5), and rurality of residence (urban/rural). The world standard population was used to perform age standardization. Incidence rate ratios (IRR) were calculated globally and in each age group to compare BC incidence between socioeconomic groups (independently, quintiles from Q2 to Q5 vs. Q1). IRR were estimated fitting multivariate Poisson regression models, adjusting for age (age classes within age groups: 15–29, 30–39, 40–49, 50–59, 60–64, 65–74, and 75+ y/o) and rurality of residence (adjustment was done both in overall and stratified analysis). Nested random effects at the municipality/IRIS level were also included in the model to account for the geographical variability of different factors which might be related to BC incidence by stage at diagnosis (in addition to socioeconomic deprivation, age, and rurality of residence). The municipality/IRIS level was chosen as the best random effect (between department, municipality, IRIS, department/municipality, and department/IRIS) by comparing the goodness of fit of the possible models with a likelihood test ratio. Once the IRRs were computed, the statistical significance of their social gradient was tested by running an analysis of variance (ANOVA). Finally, interactions between F-EDI and the urban/rural residence were tested.

Statistical analyses were performed using R software (version 4.2.0).

## 3. Results

### 3.1. Population Characteristics

The characteristics of the 33,298 women included in the study are presented in Table 1. At diagnosis, more than half of the women were aged between 50 and 74 y/o (58%), over three quarters lived in urban areas (79%), and their distribution among the F-EDI quintiles was as follows: 24% lived in the most affluent areas (i.e., Q1), 21% in Q2, 19% in Q3 as well as in Q4, and 16% in the most deprived areas (i.e., Q5). As socioeconomic deprivation increased, the proportion of women aged 75+ y/o increased (from 15% in Q1 to 23% in Q5, *p* < 0.001) as well as those living in urban areas (from 74% in Q1 to 98% in Q5, *p* < 0.001).

Most BC were diagnosed at stage I (51%), followed by stage II (31%), and stage III-IV (17%). Stage I BC was more frequent among women living in the most affluent areas (from 54% in Q1 to 49% in Q5, *p* < 0.001), while stage III-IV BC was more frequent in women living in the most deprived ones (from 14% in Q1 to 20% in Q5, *p* < 0.001). No major difference between F-EDI quintiles was observed in the distribution of stage II BC. Diagnosis of BC was mainly made by symptoms, regardless of F-EDI quintiles (40%), except for women aged 50–74 y/o for which organized screening represented the most frequent mode of detection (53%, followed by symptoms (27%) and opportunistic screening (14%)). No significant difference (*p* = 0.677) in the mode of detection was found between quintiles in this age group.

### 3.2. Age-Standardized Incidence Rates

Figure 1 provides a graphical representation of ASIR per 100,000 women-years according to age group, stage at diagnosis, and F-EDI quintile (see Appendix A reporting ASIR and 95% CIs). When considering all women, ASIR per 100,000 progressively decreased with socioeconomic deprivation for all-stages, ranging from 145.9 (95% CI 141.9–150.0) in Q1 to 124.5 (95% CI 120.6–128.4) in Q5, for stage I, stretching from 79.5 (95% CI 76.6–82.5) in Q1 to 63.9 (95% CI 61.1–66.7) in Q5, and for stage II, going from 46.7 (95% CI 44.4–49.1) in Q1 to 39.1 (95% CI 36.9–41.4) in Q5. For stages III–IV, no clear pattern was reported but ASIR differed significantly between the two extreme quintiles (19.6 (95% CI 18.2–21.1) in Q1, 21.4 (95% CI 19.8–23) in Q5).

After stratification by age group, this decreasing trend persisted for all-stages, stage I and stage II, although it was less clear in older women (75+ y/o) for all-stages and for stage I. In terms of stages III-IV, ASIR increased along with social deprivation in women aged 50 years and older, from 39.6 (95% CI 37.8–41.4) in Q1 to 45.5 (95% CI 43.1–47.9) in Q5 for women aged 50–74 y/o, and from 74.9 (95% CI 74.1–75.8) in Q1 to 94.2 (95% CI 93.2–95.1) in Q5 for those aged 75 years and older. The trend was less clear in women aged 15–49 y/o, going from 10.1 (95% CI 8.7–11.4) in Q1 to 9.6 (95% CI 8.4–10.9) in Q5.

When analyzing the urban/rural context, a lower all-stages ASIR was recorded among women aged 75+ y/o living in rural areas, while no major differences were observed for women aged < 75 y/o and in the 15+ y/o analysis (Appendix A). When considering only stage I, a lower ASIR in rural areas was found in global and age-stratified analyses, especially in women aged 50–74 y/o and 75+ y/o. On the other hand, an inverted trend (higher ASIR in rural areas) was observed for stage II (apart from women aged 75+ y/o) and for stages III–IV. In the last case, this trend was more pronounced in the organized screening target group (50–74 y/o), being 46.4 (95% CI 44.3–48.5) in rural areas vs. 41.8 (95% CI 40.7–42.8) in urban areas.

### 3.3. Incidence Rate Ratios

As represented in Figure 2 and Appendix A, IRR significantly decreased as social deprivation increased in women younger than 75 y/o for all-stages (from Q1 to Q5: −15% for 15+ y/o, −21% for 15–49 y/o, −14% for 50–74 y/o), for stage I (from Q1 to Q5: −23% for 15+ y/o, −27% for 15–49 y/o, −22% for 50–74 y/o), and for stage II (from Q1 to Q5: −16% for 15+ y/o, −20% for 15–49 y/o, −11% for 50–74 y/o). On the other hand, for stages III–IV, IRR significantly increased with social deprivation (from Q1 to Q5: +18% for 15+ y/o, +21% for 50–74 y/o), except for women aged 15–49 y/o (*p* = 0.439). Concerning the elderly group (75+ y/o), no significant deprivation trend was found, except for stage II (from Q1 to Q5: −22%).

Concurrently, all-stages IRR significantly decreased in rural areas in all women and in women aged 15–49 y/o (respectively: from Q1 to Q5 −4%, −6%) (Appendix A). The results were at the limit of statistical significance in women aged 75+ y/o (from Q1 to Q5 −7%, *p* = 0.057) and not significant for the screening target group. Analyses by stage showed significantly decreased IRR for stage I (from Q1 to Q5: −11% for 15+ y/o, −15% for 15–49 y/o, −9% for 50–74 y/o, −19% for 75+ y/o), while no significant difference was observed between rural and urban areas for stage II, and for stages III–IV in women aged 15–49 and 75+ y/o. Nevertheless, for stages III–IV, a significant increase in IRR has been observed in rural areas in 15+ y/o analyses and in the 50–74 y/o age group (respectively: from Q1 to Q5 +11%, +14%).

## 4. Discussion

Our study highlighted a significant effect of socioeconomic deprivation on invasive BC incidence in France. The direction of this social gradient changed based on the stage at diagnosis. On one hand, as socioeconomic deprivation increased, a significant reduction of incidence was observed for all-stages (−15% from Q1 to Q5) and early BC (from Q1 to Q5: stage I: −23%, stage II: −16%). On the other hand, this social gradient was inverted for advanced BC (stages III–IV), where incidence significantly increased along with socioeconomic deprivation (+18% from Q1 to Q5). These patterns persisted even after analysis stratification by age group, although was not significant in women aged 75+ y/o for all-stages, stage I and stages III-IV, and except in women aged 15–49 y/o for stages III–IV.

The overall increased BC incidence in affluent areas is coherent with previous findings reported in different European countries [1,18,19,20]. In the specific case of France, the gap between socioeconomic groups has been described as continuing to widen over time, with women from affluent areas presenting significantly higher BC incidence rates [1]. A study from Denmark revealed that increased BC incidence observed among women aged 50–69 y/o living in affluent socioeconomic neighborhoods could be partly explained by screening attendance [21]. Moreover, a positive association between screening uptake and area-based socioeconomic affluence has been documented in different European countries, including France [4,22]. Nevertheless, in age-stratified analyses, we reported a significantly higher BC incidence in affluent areas not only in women aged 50–74 y/o, but also in women aged 15–49 y/o, suggesting that the national organized screening program is not the sole factor influencing this phenomenon. As a matter of fact, the causes of BC are multifactorial and are unevenly distributed among socioeconomic groups. A European study carried out at the individual level has shown that the association between educational attainment and invasive BC risk disappeared after the adjustment by known BC risk factors, such as reproductive history (age at first full-term pregnancy, parity, age at menarche, contraceptive use, breastfeeding, menopausal status, age at menopause, and hormonal replacement therapy), lifestyle (physical activity and alcohol consumption), and anthropometric factors (height and BMI) [3]. Among those factors, age at first full-term pregnancy (higher in highly educated women), parity (lower in highly educated women), and height (higher in highly-educated women) were the ones that could explain most of the differences observed between education groups in BC incidence. Finally, overdiagnosis related to screening could represent another factor influencing incidence according to socioeconomic status, even in the younger part of the population (<50 y/o), with opportunistic screening [9].

After stratifying the analyses by stage at diagnosis, the direction of the social gradient (higher incidence in the most affluent areas) for all-stages was mainly determined by early BC (stage I and stage II), which accounted for 82% of all cancers included in the study, while the social gradient was inverted (higher incidence in the most deprived areas) for advanced BC (stages III–IV). These results are coherent with those issued from other European studies comparing stage-specific BC incidence according to socioeconomic status, despite the fact that they focused on more restricted age groups (30–48 and >49 y/o) [8,9]. On one hand, the higher incidence of early BC (stage I and II) in affluent areas could be the result of co-exposure to some of the previously reported BC risk factors and of the higher screening uptake (both organized and opportunistic screening) recorded in these areas. On the other hand, the higher advanced BC (stages III–IV) incidence rates observed in deprived areas could be explained by the increased difficulty in accessing healthcare and lower awareness of the disease, leading to lower screening uptake. For instance, two studies carried out in the Netherlands and in the Flanders region have shown that stages III–IV BC incidence in non-screened women (eligible for the organized national screening program) was more than double than in their screened counterparts [8,23]. Different factors, such as limited knowledge about mammography screening programs, misconception regarding cancer and mammography, and awareness of cancer warning signs have been identified as having negatively influenced screening uptake in deprived areas [24,25,26]. A French study has underlined the association between difficulties in access to healthcare, in particular absence of a regular follow-up by a gynecologist or by a general practitioner, with low screening uptake [27]. These findings are in line with our study and, more specifically, with the results issued from age-stratified analyses that show a significantly higher IRR (1.21 95% CI 1.06–1.38) only for women aged 50–74 y/o (screening target group) living in the most deprived areas (Q5) when compared to the most affluent areas (Q1).

The higher incidence rates of advanced BC could be a plausible main driver of the concomitant similar or higher mortality rates reported by previous studies in deprived socioeconomic neighborhoods [28,29]. Firstly, the relative 5-year survival rate is strongly associated with stage at diagnosis, decreasing from 99% for localized invasive BC to 86% for regional BC, and to 29% for distant BC [10]. Next, the excess BC lethality (+134%) in deprived areas of France has been identified as the main driver of the higher mortality (+19%) recorded in this group [28]. However, a more advanced stage at diagnosis is probably not the sole factor influencing BC survival rates in deprived areas. Differences in care related to comorbidities and unhealthy behaviors may be other elements capable of explaining BC survival disparities between socioeconomic groups [30,31,32].

In our study, we also identified an effect of the urban/rural residence on BC incidence for all-stages, stage I, and stages III-IV. When compared to urban areas, significantly decreased incidence rates were observed in rural areas for all-stages (−4%) and stage I (−11%), while a significantly increased incidence rate was reported in rural areas for stages III–IV (+11%). Age-stratified analyses presented the same results, except for all-stages, where the IRR significantly decreased in rural areas only for people aged 15–49 y/o (−6%), and for stages III–IV, only women aged 50–74 y/o living in rural municipalities presented a significantly increased IRR (+14%). These results are consistent with previous studies describing a higher all-stage BC incidence rate in urban areas [17]. Nevertheless, later stage at diagnosis and lower survival rates have been observed among women living in rural areas all across Europe [33,34,35,36]. Compared to women living in socioeconomically deprived areas, those from rural municipalities encountered more difficulties in accessing healthcare facilities (for different reasons, e.g., longer distance to travel, social environment, and lacking access to adequate health information) and have presented a lower screening uptake [22,37,38].

The results presented in this study must be interpreted taking the following limitations into consideration. Despite consistent results in cancer research between studies adopting either individual or ecological measures of socioeconomic deprivation, the use of an ecological index introduced the risk of underestimating the effect of the studied health determinant [2,39]. Moreover, the F-EDI used in this study was computed in 2011, while the BC cases were diagnosed between 2008 and 2015, introducing the risk of not capturing potential variations in deprivation during those years. This limitation was due to the lack of annually updated F-EDI data. Finally, in order to increase statistical power, we had to group stage III and stage IV BC under the same category, providing less detailed results about the effect of socioeconomic deprivation on BC incidence by stage at diagnosis. However, to our knowledge, this is the first study exploring the social gradient of invasive BC incidence in France using this degree of granularity in the stratification of analyses by stage at diagnosis. This was possible thanks to the use of high-quality data extracted from population-based registries, exhaustively recording all new BC cases in more than a tenth of the mainland French population (11.1%).

## 5. Conclusions

In conclusion, this study provides a possible explanation of the so-called BC social paradox, which consists of higher mortality and lower incidence rates in disadvantaged socioeconomic groups when compared to the most affluent groups. As a matter of fact, the analysis by stage at diagnosis revealed that (i) the overall lower incidence observed in disadvantaged groups was due to a significantly lower early (stage I and stage II) invasive BC incidence, and that (ii) a higher advanced (stage III–IV) invasive BC incidence existed among people living in deprived areas when compared to the most affluent areas. These two points could explain the higher mortality rates observed in deprived areas of France, as a later diagnosis is related to a higher case fatality rate. Reasons for this phenomenon may be found in socioeconomic inequalities in the exposure to BC risk factors, awareness, and in access to and use of healthcare services, including screening, which are underutilized in France despite the universal health coverage system and the free national BC screening program. Targeted interventions in deprived areas aimed at increasing knowledge about BC, preventable risk factors, as well as removing barriers to access to healthcare services (e.g., general practitioners, gynecologists, and, thus, screening), could be promising solutions in reducing the socioeconomic inequalities observed in BC.

## Figures and Tables

**Figure 1 cancers-16-01701-f001:**
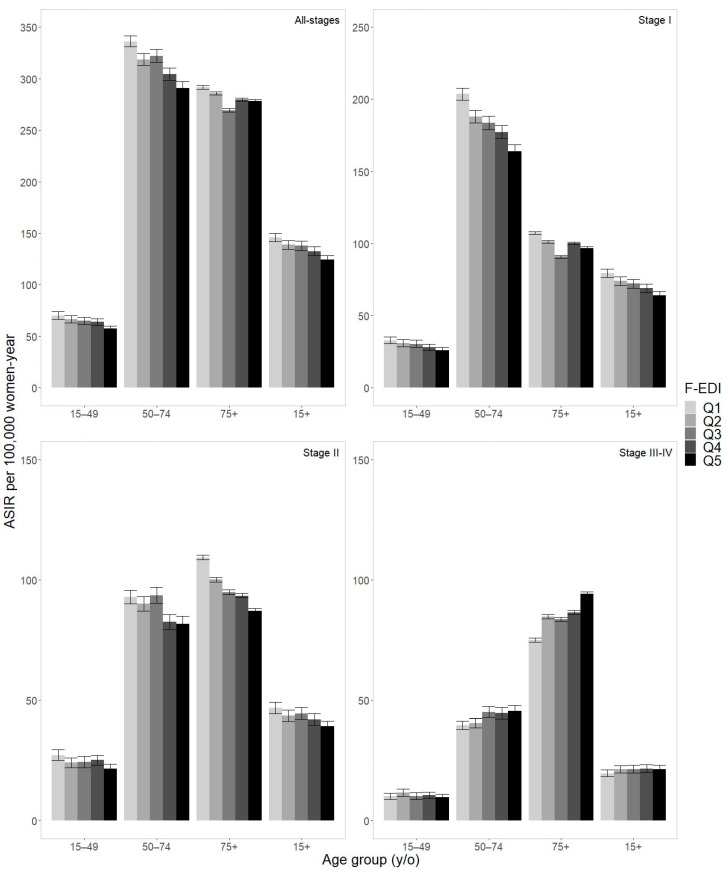
Age-standardized incidence rates (ASIR) of breast cancer and 95% confidence intervals according to age group and socioeconomic deprivation (F-EDI quintile: from Q1 the most affluent to Q5 the most deprived), by stage at diagnosis (*n* = 33,298; 2008–2015).

**Figure 2 cancers-16-01701-f002:**
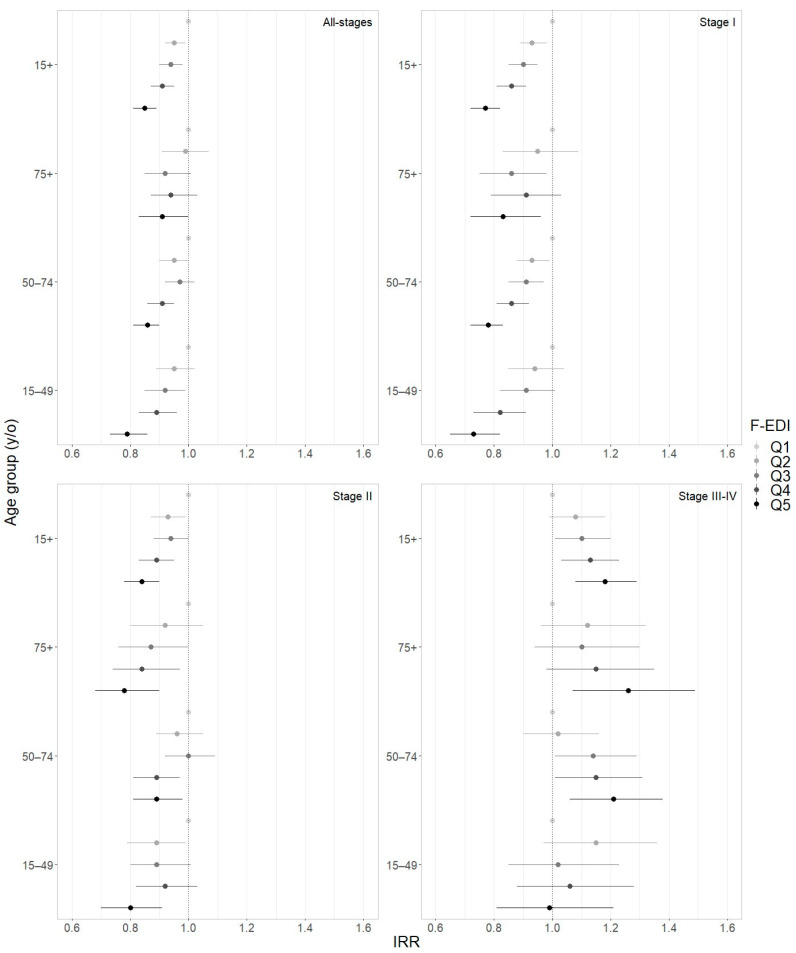
Incidence rate ratios (IRRs) and 95% confidence intervals according to age group and socioeconomic deprivation (F-EDI quintile: from Q1 the most affluent to Q5 the most deprived), by stage at diagnosis (*n* = 33,298; 2008–2015).

**Table 1 cancers-16-01701-t001:** Characteristics of women included in the study according to socioeconomic deprivation (2008–2015).

Socioeconomic Deprivation (F-EDI Quintile)	Q1 The Most Affluent	Q2	Q3	Q4	Q5 The Most Deprived		All Women
	N (%)	N (%)	N (%)	N (%)	N (%)	*p*	N (%)
**Number of women**	8143	7108	6303	6277	5467	-	33,298
**Age group** (y/o)							
15–49	2018 (25%)	1661 (23%)	1338 (21%)	1318 (21%)	1148 (21%)	<0.001	7483 (22%)
50–74	4911 (60%)	4119 (58%)	3702 (59%)	3545 (56%)	3055 (56%)		19,332 (58%)
75+	1214 (15%)	1328 (19%)	1263 (20%)	1414 (23%)	1264 (23%)		6483 (20%)
**Rurality of residence**							
Rural	2141 (26%)	2279 (32%)	1711 (27%)	871 (14%)	134 (2%)	<0.001	7136 (21%)
Urban	6002 (74%)	4829 (68%)	4592 (73%)	5406 (86%)	5333 (98%)		26,162 (79%)
**Stage at diagnosis**							
I	4380 (54%)	3685 (52%)	3178 (50%)	3158 (50%)	2687 (49%)	<0.001	17,088 (51%)
II	2586 (32%)	2217 (31%)	2004 (32%)	1941 (31%)	1678 (31%)		10,426 (31%)
III–IV	1177 (14%)	1206 (17%)	1121 (18%)	1178 (19%)	1102 (20%)		5784 (17%)
**Mode of detection**							
Symptoms	3092 (38%)	2972 (42%)	2580 (41%)	2523 (40%)	2123 (39%)	<0.001	13,290 (40%)
Organised screening	2673 (33%)	2253 (32%)	2015 (32%)	1895 (30%)	1648 (30%)		10,484 (31%)
Opportunistic screening	1368 (17%)	1190 (17%)	1100 (17%)	1111 (18%)	1030 (19%)		5799 (17%)
Other	155 (2%)	127 (2%)	130 (2%)	142 (2%)	112 (2%)		666 (2%)
Unknown	855 (11%)	566 (8%)	478 (8%)	606 (10%)	554 (10%)		3059 (9%)
**Department of residence**							
Côte d’Or	939 (12%)	581 (8%)	489 (8%)	643 (10%)	463 (8%)	<0.001	3115 (9%)
Doubs	566 (7%)	505 (7%)	571 (9%)	576 (9%)	568 (10%)		2786 (8%)
Hérault	873 (11%)	1080 (15%)	1338 (21%)	1906 (30%)	2074 (38%)		7271 (22%)
Isère	2236 (27%)	1167 (16%)	1084 (17%)	1393 (22%)	1434 (26%)		7314 (22%)
Loire-Atlantique	2843 (35%)	2174 (31%)	1493 (24%)	1129 (18%)	784 (14%)		8423 (25%)
Vendée	686 (8%)	1601 (23%)	1328 (21%)	630 (10%)	144 (3%)		4389 (13%)
**Year of diagnosis**							
2008	961 (12%)	834 (12%)	718 (11%)	727 (12%)	665 (12%)	0.879	3905 (12%)
2009	881 (11%)	824 (12%)	702 (11%)	740 (12%)	626 (11%)		3773 (11%)
2010	958 (12%)	831 (12%)	744 (12%)	768 (12%)	673 (12%)		3974 (12%)
2011	1061 (13%)	880 (12%)	790 (13%)	739 (12%)	693 (13%)		4163 (13%)
2012	1024 (13%)	928 (13%)	813 (13%)	793 (13%)	684 (13%)		4242 (13%)
2013	1069 (13%)	916 (13%)	820 (13%)	795 (13%)	705 (13%)		4305 (13%)
2014	1058 (13%)	927 (13%)	873 (14%)	856 (14%)	719 (13%)		4433 (13%)
2015	1131 (14%)	968 (14%)	843 (13%)	859 (14%)	702 (13%)		4503 (14%)

## Data Availability

The raw data supporting the conclusions of this article will be made available by the authors on reasonable request.

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
