# Peer review of "Socioeconomic Deprivation and Invasive Breast Cancer Incidence by Stage at Diagnosis: A Possible Explanation to the Breast Cancer Social Paradox"

_cancers, 2024, doi:10.3390/cancers16091701_

Round 1
Reviewer 1 Report
Comments and Suggestions for Authors
The present paper evaluates socioeconomic and urban-rural inequalities in incidence of breast cancer, considering the stage at diagnosis, in France during 2008-2015.
Major concerns
· Some terms regarding deprivation, like “deprived women”, should be replaced for their ecological equivalence, like “women living in most deprived areas”. This is written in both the abstract and the results section.
· You justify the use of data of six regions of France because it’s a “better representation of French socioeconomic deprivation”. I would like to see a map with the regions highlighted within France, and see the spatial distribution of F-EDI to check if this is true.
· There are some doubts when describing how did you use F-EDI in the material and methods section. Did you have to geocode the addresses of the cases? Were there any cases lost in this process? And how did you compute the quintiles of F-EDI? Did you use the whole country as a reference to compute the quintiles, or only the six regions? Did you consider the population of each IRIS? This additional information should be in the “Data” section to better understand the method.
· Reading the last paragraph of the “Data” section, it would be interesting to know the distribution of unknown stage by F-EDI, and the distribution of unknown F-EDI by stage, because 4.1% of all the cases is a relevant percentage of excluded cases, and there could be some relevant selection bias.
· In Figure 1, Figure 2, and Supplementary Table 1 there are some important confusions that need to be corrected. In both Figures, for stages III and IV, the ASIR of all ages is less than the ASIR for any age group, which doesn’t make sense. In Supplementary Table 1, the ASIR for “All areas” and “Stage III-IV” are all incorrect. After a very careful reading, maybe the ASIR you reported for 15-49 is actually the ASIR for All ages, the ASIR reported for all ages is the ASIR for 75+, the ASIR reported for 75+ is the ASIR for 50-74, and the ASIR reported for 50-74 is the ASIR for 15-49? And finally, if I understood correctly, the ASIR for all-stages should be the sum of the ASIR for the different stages, and these numbers are not the same in any of the categories. Please check carefully the analysis made for the figures and the Supplementary table 1 because this is the core of the manuscript.
Minor concerns
· In line 74, you referred to stage greater than 2 as “advanced”, but later (e.g. line 244) you defined advanced as stages III and IV. The latter definition is the most adequate internationally.
· In the section of “Statistical analysis”, you didn’t mention any analysis made in table 1. The p-value is the result of a chi-square? Please consider reporting the p-value of trend.
· When you estimate IRR and adjust by age, why do you not consider 75+ years old women?
· To improve readability in table 1, please consider using thousands separator (19,332 instead of 19332) and the percentage symbol.
· In table 1, it’s confusing to me why only 2% of the cases in most deprived areas are rural, and 98% urban. Are rural areas in France very affluent? The map of F-EDI suggested before could explain this, but please consider discussing this result.
· Please consider adding year of diagnosis and region to table 1, because these are some basic characteristics of the patients.
· Also in table 1, there is a category of “Other” mode of detection. Could you be more specific or better describe this category? Are these accidental diagnoses?
· In line 180, please add “women” after “per 100,000”.
· In all the figures and tables, when you state “All ages”, are you considering also women 0-14 years old? If not, better state “15+ years old”.
· In Figures 1 and 2, add “women” to the title of the vertical axis.
· In Supplementary Table 1, please use “.” as the decimal point. And also, report one decimal for each data (e.g. “30.0” instead of “30”).
· In the second paragraph of the discussion, consider citing this article of Cancers that describes, in a neighbouring country like Spain, higher breast cancer incidence in women living in more affluent areas https://www.mdpi.com/2072-6694/13/11/2820
· In the discussion, I would add as a limitation the fact that F-EDI refers to 2011 and the incidence data is from 2008 to 2015. You can explain how this is not something you can solve due to the F-EDI only updating every 10 years.
· In line 295, there is a “26” that maybe is a reference not well cited.
· In line 310, you state “stages III-V”. I believe this is a mistake.
· In line 317 you cite [16], a reference that does not relate to the statement “These results are consistent with previous studies describing a higher all-stages BC incidence rate in urban areas [16]”.
· In line 337, please consider mentioning the paradox as the “social” paradox (as it appears in the title).
· As the focus of the article is socioeconomic inequalities using F-EDI, please consider moving the figures about urban vs. rural inequalities (Figures 2 and 4) as Supplementary Figures.
Comments on the Quality of English Language· In the line 55 there is a typo: “50-74-year-old” should be “50-74 year-old”.
· When naming the French departments there are commas but also semicolons (lines 92-93).
· In line 129, “lower BC incidence rate” should be “lower BC incidence rates”.
· In Supplementary Table 2, the reference category is marked with “(réf)” which can be “référence” in French. Please replace it with “reference”.
Reviewer 2 Report
Comments and Suggestions for Authors
Borgi et al. used the invasive breast cancer data from six mainland French departments and evaluated the impact of socioeconomic status (represented by ecological F-EDI) on breast cancer incidence stratified by stage. They discovered that all-stages and early (stage I and stage II) invasive breast cancer incidence significantly decreased with increasing deprivation, while advanced (stage III-IV) breast cancer incidence significantly increased with increasing deprivation. Thus they provide an explanation on why the deprived areas have lower breast cancer incidence but yet similar or higher mortality. The data was clearly presented, and the results support the conclusion. Limitations are well outlined.
Author Response
Dear reviewer,
The authors are deeply thankful for your consideration and for the time to review this manuscript.
Best Regards,
Dr Giulio Borghi
Reviewer 3 Report
Comments and Suggestions for Authors
The presented study assessed the influence of area-based socioeconomic deprivation on the incidence of invasive breast cancer in France. The study included 33,298 patients diagnosed with invasive breast carcinoma in six French departments between 2008 and 2015.
The authors stratified the patients by stage at diagnosis, and found the higher incidence in the most affluent areas mainly composed of early breast cancer (stage I and stage II) that could partly be explained by screening attendance. On the other hand, the higher incidence of advanced breast cancer (stages III-IV) was observed in deprived areas that could be explained by the increased difficulty in accessing healthcare including screening and lower awareness of the disease. The authors provided a possible explanation of the so-called breast cancer paradox, which consists of lower incidence but higher mortality of breast cancer in disadvantaged areas primarily due to a significant lower early (stage I and stage II) invasive breast cancer diagnosis in deprived areas as a later diagnosis is related to a higher mortality rate.
The subject of this paper is of great importance since there is a necessity for increasing awareness about breast cancer and providing accessibility to healthcare and screening especially in most deprived areas.
The study is well organized, the results are interesting, the discussion is well written. The reference list covers the relevant literature with recently published articles.
I have no remarks to proposed article.
Author Response

(The authors gave the same response as above.)
